# Pacing and Performance Analysis of the World’s Fastest Female Ultra-Triathlete in 5x and 10x Ironman

**DOI:** 10.3390/ijerph17051543

**Published:** 2020-02-27

**Authors:** Caio Victor Sousa, Pantelis T. Nikolaidis, Vicente Javier Clemente-Suárez, Thomas Rosemann, Beat Knechtle

**Affiliations:** 1Bouve College of Health Sciences, Northeastern University, 360 Huntington Ave, Boston, MA 02115, USA; cvsousa89@gmail.com; 2Exercise Physiology Laboratory, Thermopylon 7, 18450 Nikaia, Greece; pademil@hotmail.com; 3Faculty of Sport Sciences, European University of Madrid, C/ Tajo, s/n. Urb. El Bosque, 28670 Villaviciosa de Odón, Spain; vicentejavier.clemente@universidadeuropea.es; 4Grupo de Investigación en Cultura, Educación y Sociedad, Universidad de la Costa, Barranquilla 080002, Colombia; 5Institute of Primary Care, University of Zurich, Pestalozzistr. 24, CH-8001 Zurich, Switzerland; thomas.rosemann@usz.ch; 6Medbase St. Gallen Am Vadianplatz, Vadianstrasse 26, 9001 St. Gallen, Switzerland

**Keywords:** cycling, fatigue, running, swimming, ultra-endurance

## Abstract

The aim of the present case study was to analyse the performance data of the world’s best female ultra-triathlete setting a new world record in a Quintuple (5xIronman) and Deca Iron (10xIronman) ultra-triathlon, within and between race days, and between disciplines (cycling and running) and races (Quintuple and Deca Iron ultra-triathlon). The subject was an elite female triathlete (52 kg, 169 cm) born in 1983. At the time of her world record in Quintuple Iron ultra-triathlon she had an age of 35 years and at the time of the world record in Deca Iron ultra-triathlon 36 years old. The distribution of time spent in each discipline and transitions was 8.48% in swimming, 51.67% cycling, 37.91% running, and 1.94% transitions. There was no difference between the race days of the average speed neither in cycling nor running. The running pace had a within-day variation larger than the cycling pace, and also varied more between race days. In conclusion, the world’s best female ultra-triathlete adopted a steady (even) pacing strategy for both cycling and running, without substantial variations within- or between race days, for both the world record in a Quintuple and a Deca Iron ultra-triathlon.

## 1. Introduction

Women started to participate in ultra-endurance triathlons in 1988, and ever since with a very low number of participants [1]. Ultra-triathlons are race events including distances beyond the traditional Ironman distance (3.8 km swimming, 180 km cycling and 42.195 km running) [2]. Popular multi-day types of ultra-triathlon have taken place since 1985, where the athletes have to perform two to ten times the Ironman distance [1]. The large physical demands of extreme ultra-endurance events preclude large participations [3].

Women are slower than men in swimming, cycling, and running in ultra-triathlons [1]. Moreover, women have less participants than men from Double Iron ultra-triathlon (2x the Ironman distance) to Deca Iron ultra-triathlon (10x the Ironman distance). But despite their low participation numbers in comparison to men, the participation increases in a higher rate than men in all endurance events throughout the years. Moreover, women’s endurance performance is also increasing, closing the performance gap to men in many endurance events, such as open-water swimming [4], Olympic distance triathlon [5], Ironman triathlon [6], and also ultra-triathlons [1].

Regarding to endurance performance, there is evidence that women have less fatigability than men [7]. Pacing strategy is another aspect that could affect women’s performance in ultra-endurance events [8,9]. Different pacing strategies, such as negative, positive, or steady pacing may be closely related to performance in endurance events [8]. Nevertheless, the literature on women’s performance is still very poor. To the best of our knowledge, there are no studies that investigated the race strategy and pacing variation of a female athlete in ultra-triathlon events. 

Therefore, we analyzed the performance data of the best female ultra-triathlete setting a new world record in both a Quintuple (5xIronman) and Deca Iron (10xIronman) ultra-triathlon. The aim was to investigate the performance within and between race days, also comparing between disciplines (i.e., cycling and running) and races (i.e., Quintuple and Deca Iron ultra-triathlon). Additionally, we also aimed to determine whether the pacing variation has an association with overall performance. These results would be the first scientific report of a female ultra-triathlete performance for athletes and coaches to reference for a race strategy plan.

## 2. Materials and Methods 

### 2.1. Ethical Concerns

This study was approved by the Institutional Review Board of Kanton St. Gallen, Switzerland, with a waiver of the requirement for informed consent of the participants as the study involved the analysis of publicly available data (EKSG 01-06-2010). The athlete also gave an informed consent that all her personal data provided were going to be published. All procedures adhered to the ethical standards set by the Declaration of Helsinki.

### 2.2. Athlete Characteristics and Competition Background 

Our subject was an elite female triathlete (52 kg, 169 cm) born in 1983 in Switzerland. The athlete’s body mass remained stable during training routine for both races. At the time of her World record in Quintuple Iron ultra-triathlon she had an age of 35 years and at the time of the World record in Deca Iron ultra-triathlon 36 years old. She improved the existing world record in Quintuple Iron by more than 9 h and the existing world record in Deca Iron ultra-triathlon by more than 11 h.

She started her athletic career as short distance triathlete in 2005 at the age of 22 years where she achieved several podiums at national level (Switzerland). In 2006 (23 years) she obtained 2nd place in her age group in Ironman Lanzarote. From 2007 to 2008, she competed as elite cyclist at national level (Switzerland). Between 2009 and 2015 she made a break due to pregnancies and trained only a few hours per week. In 2016, she started competing in multi-day races with first place women in Gigathlon Switzerland, an event consisting of 5 disciplines (i.e., swimming, cycling, running, inline skating and mountain biking). In 2017, she won Gigathlon Czech Republic and obtained 2nd places in Megathlon Germany, Gigathlon Switzerland, and Biennathlon Switzerland. Also, these races consisted in several disciplines. In 2019, she won the Rocky Mountain Bike marathon in Germany and a Double Iron ultra-triathlon (7.6 km swimming, 360 km cycling, and 84.4 km running) held in Austria in a new Swiss record.

Regarding her training, she invested as triathlete 8–16 h per week with a maximum of 35 h per week before the world record races. She invested <5% of her time in swimming, ~50% in cycling, ~40% in running, and the rest in resistance training (upper body). As an elite cyclist, she trained at that time for 10–15 h per week.

### 2.3. Races Characteristics

The athlete achieved the world records at the races of the “swissultra” [10]. This race has been held annually in August since 2015 and offers 5xIronman, 10xIronman, and 20xIronman.

The Quintuple and Deca Iron ultra-triathlon consist of a race to be performed in multiple days (5 days for a Quintuple and 10 days for a Deca Iron ultra-triathlon). Each athlete has to perform an Ironman original distance on each race day (i.e. 3.8 km swimming, 180 km cycling and 42.2 km running). These races take place in closed circuits where the athlete performs several laps until the goal distance is achieved. Swimming is held in a 50-m outdoor pool with a temperature of 20−23 °C. Cycling is held on a completely flat and traffic-free course where 20 laps of 9 km must be performed. Running is held on a completely free course where 35 laps of 1.2 km must be completed. Each lap is measured electronically with a chip system. The race is held in the last two weeks of August where temperatures vary during the day from 25−35 °C. Often, rain falls in the late afternoon and evening. The cycling course is held in a large and broad valley (Rheintal) where the first half turns from north to south to a turning point to complete the second half from south to north. Early in the morning the wind is blowing from south to north and changes before noon from north to south. Each athlete had their own support for proper hydration, nutrition, and eventual mechanical issues during the race.

### 2.4. Race Strategy of the Athlete

During the Quintuple Iron ultra-triathlon, the athlete had her husband as only support. During the Deca Iron ultra-triathlon, a crew of six persons was working in shifts. The athlete and her husband had their motorhome just a few meters away from the race centre for short distances before and after each stage. Overall, she followed no specific pacing strategy. Regarding nutrition, she consumed primarily during all three disciplines liquid energy from commercial products (WOO^®^). During cycling, she ate a sandwich, and during running she drank Coca Cola and a carbohydrate-electrolyte beverage. After each stage, she ate the set meal consisting of carbohydrates, protein, and fat which was provided by the race organizer. However, she does not eat vegetables due to stomach discomforts. 

### 2.5. Data Analysis

Overall race times, split times (i.e. swimming, cycling and running) and lap times (i.e., cycling and running) were obtained from the official race website, athlete’s staff and the athlete. Data from each day of the Quintuple and Deca Iron ultra-triathlon were displayed in overall and split analyses. Data are expressed as mean, standard deviation (SD) and the coefficient of variation (%) calculated based on each lap for each race day. Additionally, a repeated measures ANOVA with within- and between-interactions was applied to detect interactions in average performance throughout the days (within; time) for cycling and running (between; group). Linear and non-linear regression were performed to investigate the trend of split and overall race times over days. The significance level was set as *p* < 0.05. Statistical Software for the Social Sciences was used in all statistical procedures (SPSS v25, Chicago, Ill, USA).

## 3. Results

Table 1 shows the split and overall race time in each day for the Quintuple Iron ultra-triathlon of the athlete. The distribution of time spent in each discipline and transitions were as follows: swimming = 8.6%; cycling 51.7%; running = 37.9%; transitions = 1.9%. Table 2 shows the split and overall race time in each day for the Deca Iron ultra-triathlon of the athlete. The distribution of time spent in each discipline and transitions were as follows: swimming = 8.5%; cycling 50.4%; running = 39.3%; transitions = 1.7%.

Figure 1 displays the average speed to complete each lap in cycling and running in each day in a Quintuple Iron ultra-triathlon completed by the female athlete. Figure 2 displays the average speed to complete each lap in cycling and running in each day in a Deca Iron ultra-triathlon completed by the female athlete. In the Deca Iron ultra-triathlon, the non-linear regression shows a negative slope for both cycling (slope = −3.52; R^2^ = 0.75) and running (slope = −0.74; R^2^ = 0.33). Time-effect was not significant neither to cycling or running in Deca Iron ultra-triathlon; and cycling was faster than running in all race days (Figure 3B). The non-linear regression shows a negative slope for cycling (slope = -1.82; R^2^ = 0.82) and close to zero for running (slope = 0.01; R^2^ = 0.81).The overall performance analysis showed a positive slope for increasing race time throughout the race days for both Quintuple (slope = 85.7; R^2^ = 0.69; Figure 3A) and Deca (slope = 21.0; R^2^ = 0.84; Figure 3B) Iron ultra-triathlon. Swimming performance analysis showed a positive slope for increasing race time throughout the race days for Quintuple (slope = 1.66.7; R^2^ = 0.43) and negative for Deca (slope = -0.43; R^2^ = 0.35) Iron ultra-triathlon.

The running started (Day 1) with the highest CV (17.1%) and had the lowest in the last day (9.0%). The average speed in Deca Iron was not significantly different along race days to neither cycling or running (Figure 4A). The coefficient of variation (CV) was similar throughout the five days for the cycling, with the highest CV in the first day (8.9%) and lowest in day 5 (5.3%).The average CV in Deca Iron was higher in running than cycling (Figure 4B). The CV was very stable (~4.5%) for the cycling throughout the ten race days. Whereas for the running it started very high (12.1%), with variations in the mid days (6 to 10%) and reached a peak in the last day (20.2%).

## 4. Discussion

This is the first case report analysing performance data of a Quintuple and a Deca Iron ultra-triathlon of a female triathlete setting in both races a new world record. The main results were (i) an even pacing during cycling and running for each stage, (ii) a positive pacing for all split disciplines and overall race time across days, (iii) no difference between the race days of the average speed neither in cycling nor running, and (iv) the running pace has a within-day variation larger than cycling, and also varies more between race days.

The present case report with the world’s fastest female ultra-triathlete shows that she applied a steady strategy (even pacing) for both running and cycling for each day. A pacing pattern can be categorized as positive (i.e., slow start, increasing speed throughout the course); negative (i.e., fast start, decreasing speed throughout the course); steady or even (i.e. close to the average throughout the course); or an irregular or specific strategy based on external factors, such as course variations, adversaries, weather [11]. Evidence shows that, in general, the best strategy to achieve the fastest possible average pace is a steady pace [12]. It is noteworthy that, in the present results, some splits were very below the average due to quick stops to reload hydration and nutrition supplies.

The finding of an even pacing during the cycling and running split in an Ironman triathlon is in contrast to existing reports about pacing of elite female and male Ironman triathletes. A study analysing 7687 cycling and 11,894 running split times of 1,392 elite Ironman triathletes (1,263 men and 129 women) showed a continuous decrease (i.e., positive pacing) in both cycling and running [13]. A potential explanation for the even pacing of this athlete could be her background as elite cyclist and her previous experience in achieving podiums or victories in similar races (e.g., Megathlon Germany, Gigathlon Switzerland, and Biennathlon Switzerland). The finding of a positive pacing across days is, however, consistent with findings of elite male triathletes competing in a Deca Iron ultra-triathlon [14,15]. In six male official finishers in a Deca Iron ultra-triathlon, split and overall race times increased linearly across the ten days [14].

Running showed a higher pacing variation than cycling. These results corroborate with previous data of men in similar races [11]. It is often reported by triathletes that ingesting solid food or even drinking, since is harder in running then cycling, and a quick reduction or quick stop (to unwrap or eat something) in running suddenly drags the speed to zero, whereas in cycling even when one stop stroking, speed drops slowly, reducing substantial variations. In other ultra-endurance running event, like a mountain ultra-endurance race the pacing pattern was negative, showing the difficulty of pace control in mountain events [16], as well as in ultra-endurance relay probes, where the rest periods could detract the real fatigue state of the athletes allowing them to show a negative pacing too [17].

The coefficient of variation in the first five days of the Deca Iron ultra-triathlon showed a similar pattern than in the Quintuple Iron ultra-triathlon in both, running and cycling. Nevertheless, at the end of the Deca Iron ultra-triathlon, the coefficient of variation in running was increased (reaching double the previous values) being cycling maintained. The lower muscular demands and the possibility of continuous food and hydration supplies, with lower assimilation problems (no vertical movements of centre of gravity) [18] could allow for a better control of pace, showing lower variations. In contrast, running limited the ingestion of both hydration and food supplies (vertical movements of centre of gravity, impact, movements) [19], fact that could modified the energy input necessary to maintain a steady pace during the race. Taking into account research in extreme environments, strenuous activities during various days could negatively affect cortical arousal, decreasing the information processing, hydration perception and rated of perceived exertion of the subjects [20]. This fact could also negatively affect operative pace control, increasing the coefficient of variation in the running segment. In this line, and even athletes in these ultra-endurance probes showed large resilience and stable psychological profile [3]. The maintenance of high motivation in such big events is an important fact that could affect the pace control and performance, but future studies should control this variable to allow to a better compression of psychological and physiological patter related with success in this extreme sport events.

Although the present case study brings valuable and novel information, it is not free of limitations. Nutritional intake reports would be practical information to be analysed and reported for future athletes. Acute physiological responses and anthropometrics during and after the race are also very important measures to assess the metabolic demand in each race day that could be related to the pacing strategy. Nonetheless, the present data it is important for athletes and coaches of such races and to give insight for new study designs to investigate the physiological responses along the race days or larger studies with a representative sample.

### Practical Applications

Considering the increased participation of female triathletes in ultra-endurance races, there has evolved a need of coaches for information on performance aspects of elite athletes. Strength and conditioning coaches working with triathletes should be aware of the relative contribution of a race’s discipline (i.e., time spent in swimming, cycling, and running) in overall race time in order to tailor training. The present case study of an elite triathlete provided a performance profile that coaches might use as a guide for their own female triathletes when planning a race strategy. 

## 5. Conclusions

The world’s best female ultra-triathlete adopted a steady (even) pacing strategy for both cycling and running, without substantial variations within- or between race days, for both the world record in a Quintuple and a Deca Iron ultra-triathlon. In summary, this case study shows that a female world record setter in a multi-day Ironman triathlon competes with an even pacing during the cycling and running splits but adopts a positive pacing over days. Most probably, such a strategy can only be achieved after long-term preparation and the experience of having completed and finished several similar races on the podium.

## Figures and Tables

**Figure 1 ijerph-17-01543-f001:**
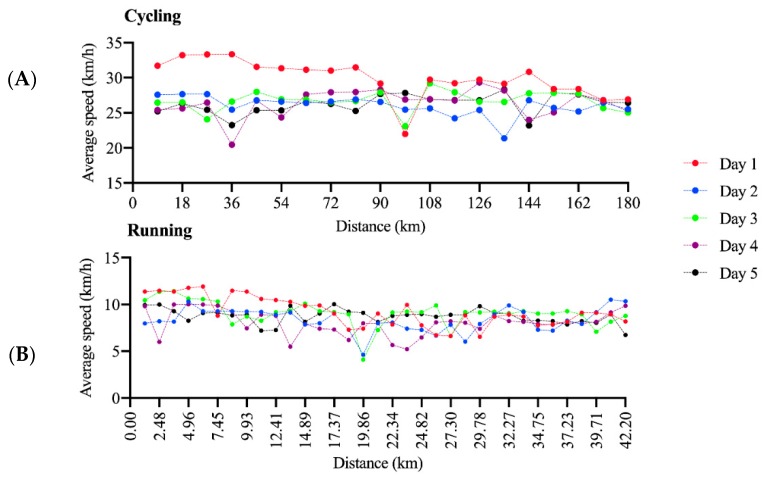
Pacing in each of the five days in a Quintuple Iron ultra-triathlon in the cycling (**A**) and running (**B**) split.

**Figure 2 ijerph-17-01543-f002:**
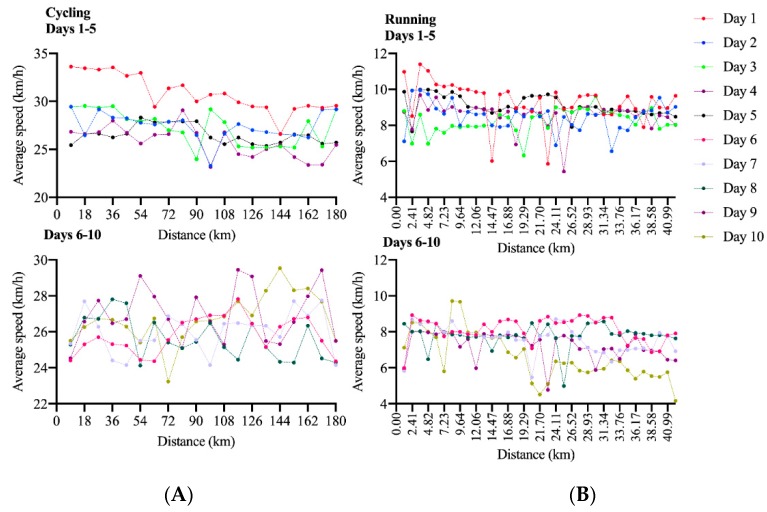
Pacing in each of the ten days in a Deca Iron ultra-triathlon in the cycling (**A**) and running (**B**) split.

**Figure 3 ijerph-17-01543-f003:**
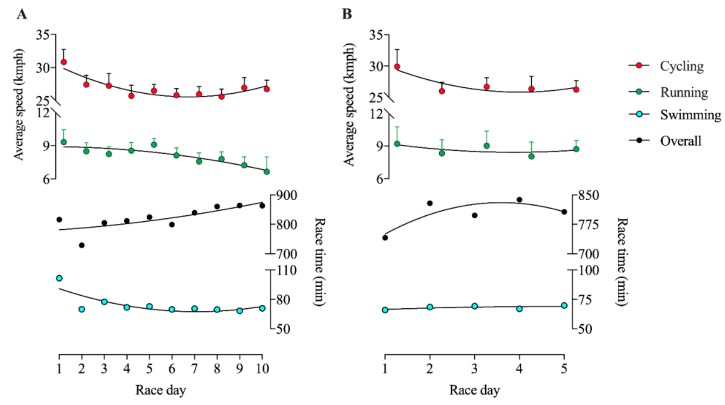
Performance (pacing, speed or race time) in each five days in a Deca (**A**) and Quintuple (**B**) Iron ultra-triathlon in the cycling and running.

**Figure 4 ijerph-17-01543-f004:**
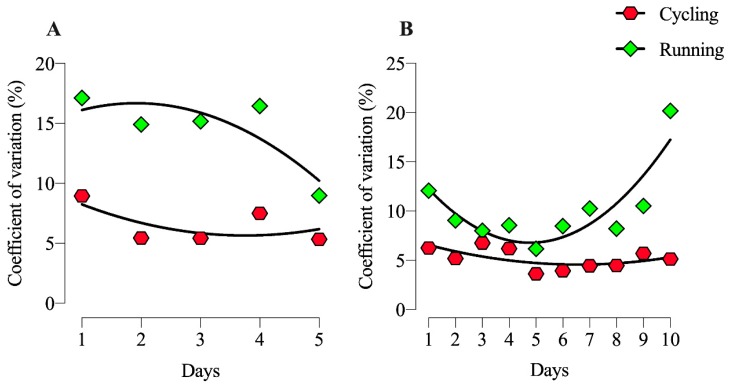
Pacing variation (coefficient of variation; CV) in each of the race days in a Quintuple (**A**) and Deca (**B**) Iron ultra-triathlon in the cycling and running.

**Table 1 ijerph-17-01543-t001:** Race time in a Quintuple Iron ultra-triathlon.

	Race time (hours : minutes : seconds)				
	Overall	Swimming	Cycling	Running	Transitions
Day 1	12:20:45	1:05:56	6:14:28	4:50:51	0:09:30
Day 2	13:48:37	1:08:25	7:09:37	5:16:29	0:14:06
Day 3	13:17:49	1:09:12	6:58:10	4:54:36	0:15:51
Day 4	13:58:01	1:07:00	7:03:47	5:30:46	0:16:28
Day 5	13:27:00	1:09:47	7:06:59	4:48:28	0:21:46
Average (SD)	13:22:26 (38:04)	1:08:04 (01:35)	6:54:36 (22:50)	5:04:14 (18:32)	0:15:32
Total	66:52:12	5:40:20	34:33:01	25:21:10	1:17:41

SD: standard deviation (minutes:seconds). Each day consists of 3.8 km of swimming, 180 km of cycling and 42.195 km of running.

**Table 2 ijerph-17-01543-t002:** Race time in a Deca Iron ultra-triathlon.

	Race time (hours : minutes : seconds)				
	Overall	Swimming	Cycling	Running	Transitions
Day 1	12:08:39	1:09:55	6:02:55	4:43:19	0:12:30
Day 2	13:24:56	1:17:28	6:50:04	5:07:27	0:09:57
Day 3	13:31:41	1:11:45	6:53:49	5:15:35	0:10:32
Day 4	13:44:39	1:12:47	7:14:34	5:05:02	0:12:16
Day 5	13:19:25	1:09:47	7:08:04	4:46:30	0:15:04
Day 6	13:59:33	1:10:35	7:09:48	5:21:21	0:17:49
Day 7	14:20:33	1:09:38	7:11:04	5:46:10	0:13:41
Day 8	14:24:21	1:08:20	7:25:29	5:35:13	0:15:19
Day 9	14:23:19	1:10:52	6:55:04	6:00:35	0:16:48
Day 10	15:03:52	1:08:24	6:55:53	6:44:24	0:15:11
Average (SD)	13:50:06 (48:16)	1:10:57 (02:40)	6:58:40 (22:31)	5:26:34 (36:42)	0:13:55 (02:35)
Total	138:20:58	11:49:31	69:46:44	54:25:36	02:19:07

SD: standard deviation (minutes:seconds). Each day consists of 3.8 km of swimming, 180 km of cycling and 42.195 km of running.

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
