# Peer review of "Pacing and Performance Analysis of the World’s Fastest Female Ultra-Triathlete in 5x and 10x Ironman"

_ijerph, 2020, doi:10.3390/ijerph17051543_

Round 1

Reviewer 1 Report

I have read the article entitled “Pacing and performance analysis of the world’s 2 fastest female ultra-triathlete in 5x and 10xIronman”. I cannot, therefore, recommend this paper for publication in International Journal of Environmental Research and Public Health in the present form.

Minor points

Line 206 Change Nerveless for Neverthless.

Homogenize references some have DOI and others do not.

The Introduction paragraph should have contained more information

Line 66 specify the country of origin of the athlete.

Figure 2 running. Look for a new way to represent the data, maybe it can be in two graphs.

Figure 3 include in the caption that means A and B.

Figure 4 identify A and B in the figure.

Major points

The conclusion of the case study does not match in the sections of abstract and conclusion.

Abstract

“In conclusion, the world’s best female ultra-triathlete adopted a steady (even) pacing strategy for both cycling and running, without substantial variations within- or between race days.”

Conclusion

In summary, this case study shows that a female world record setter in a multi-day Ironman triathlon competes with an even pacing during the cycling and running splits but adopts a positive pacing over days.

Author Response

I have read the article entitled “Pacing and performance analysis of the world’s 2 fastest female ultra-triathlete in 5x and 10xIronman”. I cannot, therefore, recommend this paper for publication in International Journal of Environmental Research and Public Health in the present form.

Minor points

Line 206 Change Nerveless for Neverthless.

Reply: changed as requested.

Homogenize references some have DOI and others do not.

Reply: changed as requested. Only the article below was not included a DOI since the journal do not have it.

Suarez, V.J.C.; Arroyo, V.E.M.F.; Campo, D.R.; Valdivielso, F.N.; Rave, J.M.G.; Santos-Garcia, D.J. Analysis of selected physiological performance determinants and muscle damage in a 24-hour ultra-endurance relay race : brief clinical report. International SportMed Journal 2011, 12, 179-186.

The Introduction paragraph should have contained more information

Reply: three paragraphs of the introduction were rephrased. Additionally, more information about the athlete were added in the methods. We believe to the reviewer’s appointment have been addressed, but if any additional information is missing, we invite the reviewer to point it out specifically and we would be help to make an effort to provide.  

Line 66 specify the country of origin of the athlete.

Reply: the athlete is from Switzerland. This information was added, as requested.  

Figure 2 running. Look for a new way to represent the data, maybe it can be in two graphs.

Reply: Figure 2 were divided into 4 plots, as suggested.

Figure 3 include in the caption that means A and B.

Reply: changed as requested.

Figure 4 identify A and B in the figure.

Reply: changed as requested.

Major points

The conclusion of the case study does not match in the sections of abstract and conclusion.

Abstract

“In conclusion, the world’s best female ultra-triathlete adopted a steady (even) pacing strategy for both cycling and running, without substantial variations within- or between race days.”

Conclusion

In summary, this case study shows that a female world record setter in a multi-day Ironman triathlon competes with an even pacing during the cycling and running splits but adopts a positive pacing over days. 

Reply: the reviewer is correct; we changed the main conclusion. Please see all changes marked as red.

Reviewer 2 Report

Dear authors, it was my pleasure to read your work regarding pacing and performance analysis of the world’s fastest female ultra-triathlete in 5x and 10x Ironman. Below are some ideas on the paper content.

INTRODUCTION

As seen in the paper, ultra-triathlon is well described as a sport discipline. Yet, less data is related to both performance and factors influencing performance. Yes, there are mentions’ regarding pacing and influence over performance but more variables could be taken into account relating even physiological aspects, specific in ultra-triathletes.

Lines 34 – 39. are quiet long and the information is lost along the way.

Lines 40 – 42. can be separated

In lines 43 – 44. the authors refer to endurance performance in women as against men, but still important differences are seen over the two groups regarding speed, resistance, strength that can overall limit performance in one group as against another

Line 48. Which are the other aspects that could affect women’s performance?

Lines 53 – 56. Are quiet long and the information is lost along the way.

Lines 59 – Authors refer to training but no data is related to volume, intensity and training periodization being unclear how the data in this paper can be used in training prescription

 METHODOLOGY

Line 64. Refers to the individual acceptance to work with public available data?

Lines 66. What kind of characteristics? History can make another chapter written on sports disciplines, national, international results

Line 67. Can the authors discuss more about the athletes’ body mass?

Line 76. Refers to several hours per week but more specific data is needed have a greater understanding of the training

Lines 83-86. Should be improved with more specific training data, regarding periods, volume and exercise intensity

In line 107. the authors state that overall, she followed no specific race or pacing strategy. Such a phrase can severely alter the results, the methodology and the working hypothesis

Lines 113 – 116. The study data was obtained directly from the official race website? I believe that the methodology needs to be improved and more data is needed.

RESULTS

The result should refer to anthropometric data, training analysis and physiological data which influenced or not the athletes pacing activity during the competition. Yet, time, distance and speed could be used more in training by athletes and coaches.

The authors’ refer to CV over speed and pace. However, no data was described regarding the actual route and the environmental conditions, which can strongly influence the pace.

How did the cycling pacing influenced the running outcome? Results regarding cycling performance and running outcomes could be illustrated.

DISCUSSIONS

Line 174 refers to an even pacing during cycling and running for each stage, while in line 177 the authors refer to running as which varies between race days.

Line 234. refers to using the study data in training guidance. How can the data be used? The authors refer to a performance profile. Yet, the performance profile often refers to oxygen consumption, power, strength, endurance, speed, exercise economy, as against pacing strategies. What kind a profile did the authors refer to?

Author Response

Dear authors, it was my pleasure to read your work regarding pacing and performance analysis of the world’s fastest female ultra-triathlete in 5x and 10x Ironman. Below are some ideas on the paper content.

INTRODUCTION

As seen in the paper, ultra-triathlon is well described as a sport discipline. Yet, less data is related to both performance and factors influencing performance. Yes, there are mentions’ regarding pacing and influence over performance but more variables could be taken into account relating even physiological aspects, specific in ultra-triathletes.

Lines 34 – 39. are quiet long and the information is lost along the way.

Reply: the paragraph was rephrased to be more objective.

Lines 40 – 42. can be separated

Reply: changed and rephrased as follows: “Women are slower than men in swim, cycling and running in ultra-triathlons [2]. Moreover, women have less participants than men from Double Iron ultra-triathlon (2x the Ironman distance) to Deca Iron ultra-triathlon (10x Ironman distance). But despite their low participation numbers in comparison to men, it increases in a higher rate.” See al changes in the manuscript marked as red.

In lines 43 – 44. the authors refer to endurance performance in women as against men, but still important differences are seen over the two groups regarding speed, resistance, strength that can overall limit performance in one group as against another

Reply: we with the expert reviewer that there still a big different between men and women in all mentioned aspects. However, in lines 43-44 we were highlighting that, despite the existing difference, we use some evidence to the show that women ARE closing the gap in some ENDURANCE sports.

Line 48. Which are the other aspects that could affect women’s performance?

Reply: rephrased as follows: “Pacing strategy is another aspect that could affect women’s performance in ultra-endurance events.” See all changes in the manuscript marked as red.

Lines 53 – 56. Are quiet long and the information is lost along the way.

Reply: the paragraph was rephrased to be clearer.

Lines 59 – Authors refer to training but no data is related to volume, intensity and training periodization being unclear how the data in this paper can be used in training prescription

Reply: we agree with the expert reviewer and removed unsuitable mentions.

 METHODOLOGY

Line 64. Refers to the individual acceptance to work with public available data?

Reply: the ethical committee approved the analysis of publicly available data from races. We add now in addition that the athlete provided personal data and gave her informed consent to use the data for publication in the case report. This information was added to the manuscript.

Lines 66. What kind of characteristics? History can make another chapter written on sports disciplines, national, international results

Reply: with “history” we meant “competition background”. The subheading was changed to avoid confusion.

Line 67. Can the authors discuss more about the athletes’ body mass?

Reply: unfortunately, the athlete does not have body mass data. Her report was that her body mass remained stable along the training and race days. This information was added to the manuscript.

Line 76. Refers to several hours per week but more specific data is needed have a greater understanding of the training

Reply: unfortunately, the athlete does not have training data. Her report was that she did not followed specific and regular training routines due to family and work.

Lines 83-86. Should be improved with more specific training data, regarding periods, volume and exercise intensity

Reply: The athlete has no training records; she trains when she has time. This is due to her duties as mother and nurse. Obviously, this training regime is of success since she is one of the best Swiss multi-day athletes and two times World record holder in multi-day triathlon. It is not always necessary to follow a strict training plan.

In line 107. the authors state that overall, she followed no specific race or pacing strategy. Such a phrase can severely alter the results, the methodology and the working hypothesis

Reply: we agree with the expert reviewer that it is confusing. What we meant is that the athlete did not followed any specific planned strategy by organizers or researchers. Since this is an observational study, we only recorded that data of interest and the athlete was free to race as she pleases.  

Lines 113 – 116. The study data was obtained directly from the official race website? I believe that the methodology needs to be improved and more data is needed.

Reply: official race data were obtained from the official results publicly available on the website. But personal information about the athlete were obtained from the athlete and athlete’s staff. This information was added, as requested.

RESULTS

The result should refer to anthropometric data, training analysis and physiological data which influenced or not the athletes pacing activity during the competition. Yet, time, distance and speed could be used more in training by athletes and coaches.

Reply: unfortunately, the athlete did not have anthropometric, training or physiological data to provide. Although we believe this would be valuable information, it was not the aim of this case report. However, we added as limitation that a secondary analysis with this data would help to better understand the results.

The authors’ refer to CV over speed and pace. However, no data was described regarding the actual route and the environmental conditions, which can strongly influence the pace.

How did the cycling pacing influenced the running outcome? Results regarding cycling performance and running outcomes could be illustrated.

Reply: information about the route and environmental conditions are described in “Race characteristics”.

DISCUSSIONS

Line 174 refers to an even pacing during cycling and running for each stage, while in line 177 the authors refer to running as which varies between race days.

Reply: at all times we were referring to pacing in both cycling and running. The rephrased the paragraph to become clearer.

Line 234. refers to using the study data in training guidance. How can the data be used?

Reply: we agree with expert reviewer that, with no training data available, this statement is not suitable. We therefore remove it.

The authors refer to a performance profile. Yet, the performance profile often refers to oxygen consumption, power, strength, endurance, speed, exercise economy, as against pacing strategies. What kind a profile did the authors refer to?

Reply: we added that the “performance profile” refers to a race strategy. Please see all changes marked as red.

Reviewer 3 Report

Overall this case study of a record-setting female ultra-distance triathlete is interesting and useful. My only big comment would be to go back through for English language corrections. I’m always impressed when people write in a second language, but nevertheless there are a number of sentences that should be tweaked for grammatical corrections. I also think it would be helpful to add some additional detail/discussion in a few places.

A few small points - 

49 - comma after positive

57-59 “These results would be the first scientific report of a female ultra-triathlete performance for athletes and coaches to reference for training prescription and race strategy plan.” I see you mentioned her training volume at lines 83-86, but I’m not sure that would allow coaches to reference very much for training prescription. Perhaps you can add some more about her training later in the discussion?

62 - The IRB approved a waiver of the requirement for informed consent because you were analyzing public data. It seems odd to read about so much personal detail in the next paragraph (lines 72-82), especially her pregnancy. Is the athlete aware of this paper?

90 - “of” instead of “in”

92 - ‘on’ instead of ‘in’

110-111 - what did that meal consist of? 

112 - Is the CV calculated based on each lap for each race? Can you explain here

121 - closed parentheses after USA

133-135  - can you re-phrase this, it is not clear what you are trying to say

Tables and figure captions - the use of the phrase ‘the world’s best female triathlete” seems odd. In the title you use the phrase “ world’s fastest female ultra-triathlete”, which is more appropriate. While I don’t think either phrase is needed for these table/figure captions, I do think some of the pro women doing 8.5 hr Ironmans might take issue with calling her ‘the world’s best female triathlete”.

146 - “The average speed in Deca Iron was different between the days to neither cycling or running” I’m unsure what you’re trying to say here?

147 - Period before “The”

166 - Figure caption should have explanation of what A and B are (like in fig 4)

194-195 - is ref 15 the correct one here?

206 - typo on “Nerveless” (should be nevertheless); also the end of this sentence isn’t clear.

210 - In contrast

213-216 - this sentence is not clear

217 - especially or specifically? (not specially)

218 - period after ref 3, then start a new sentence

226-227 - what types of studies could or should be done?

Author Response

Overall this case study of a record-setting female ultra-distance triathlete is interesting and useful. My only big comment would be to go back through for English language corrections. I’m always impressed when people write in a second language, but nevertheless there are a number of sentences that should be tweaked for grammatical corrections. I also think it would be helpful to add some additional detail/discussion in a few places.

A few small points - 

49 - comma after positive

Reply: changed as requested.

57-59 “These results would be the first scientific report of a female ultra-triathlete performance for athletes and coaches to reference for training prescription and race strategy plan.” I see you mentioned her training volume at lines 83-86, but I’m not sure that would allow coaches to reference very much for training prescription. Perhaps you can add some more about her training later in the discussion?

Reply: The athlete has no training records; she trains when she has time. This is due to her duties as mother and nurse. Obviously, this training regime is of success since she is one of the best Swiss multi-day athletes and two times World record holder in multi-day triathlon. It is not always necessary to follow a strict training plan.

62 - The IRB approved a waiver of the requirement for informed consent because you were analyzing public data. It seems odd to read about so much personal detail in the next paragraph (lines 72-82), especially her pregnancy. Is the athlete aware of this paper?

Reply: The personal data of the athlete are available on her website. Furthermore, she gave written informed consent that all her personal data she provided to us were published in her case report. This information was added in the article.

90 - “of” instead of “in”

Reply: changed as requested.

92 - ‘on’ instead of ‘in’

Reply: changed as requested.

110-111 - what did that meal consist of? 

Reply: the information was added, as suggested.  

112 - Is the CV calculated based on each lap for each race? Can you explain here

Reply: The CV was calculated based on each lap for each race day. This information was added, as requested.

121 - closed parentheses after USA

Reply: changed as requested.

133-135  - can you re-phrase this, it is not clear what you are trying to say

Reply: the changes were made as follows: “Time-effect was not significant neither to cycling or running in Deca Iron ultra-triathlon; and cycling was faster than running in all race days (Figure 3-B).” See all changes in the manuscript marked as red.

Tables and figure captions - the use of the phrase ‘the world’s best female triathlete” seems odd. In the title you use the phrase “ world’s fastest female ultra-triathlete”, which is more appropriate. While I don’t think either phrase is needed for these table/figure captions, I do think some of the pro women doing 8.5 hr Ironmans might take issue with calling her ‘the world’s best female triathlete”.

Reply: we agree with expert reviewer and adapted the captions. Although we kept in the title since we specify the race (5x 10x ultra Iron). Please be aware that no female pro athlete will be able to do again an Ironman after a 8.5 hr Ironman the next morning!  

146 - “The average speed in Deca Iron was different between the days to neither cycling or running” I’m unsure what you’re trying to say here?

Reply: the sentence was rephrased as follows: “The average speed in Deca Iron was not significantly different along the days to neither cycling or running (Figure 4-A).” Please see all changes marked as red.  

147 - Period before “The”

Reply: changed as requested.

166 - Figure caption should have explanation of what A and B are (like in fig 4)

Reply: captions were adapted as requested.

194-195 - is ref 15 the correct one here?

Reply: The correct reference is 14. It was changed.  

206 - typo on “Nerveless” (should be nevertheless); also the end of this sentence isn’t clear.

Reply: changed as requested.

210 - In contrast

Reply: changed as requested.

213-216 - this sentence is not clear

Reply: the sentence was changed as follows: “Taking into account research in extreme environments, strenuous activities during various days could negatively affect cortical arousal, decreasing the information processing, hydration perception and rated of perceived exertion of the subjects.” See all changes in the manuscript marked as red.

217 - especially or specifically? (not specially)

Reply: we just removed the word.

218 - period after ref 3, then start a new sentence

Reply: changed as requested.

226-227 - what types of studies could or should be done?

Reply: the information was added as follows: “Nonetheless, the present data it is important for athletes and coaches of such races and to give insight for new studies designs to investigate the physiological responses along the race days or larger studies with a representative sample.” See all changes in the manuscript marked as red.

Round 2

Reviewer 1 Report

The authors have corrected the manuscript in accordance with almost the recommendations. The manuscript could be accepted in its revised form.